# Intelligent Fault Diagnosis Techniques Applied to an Offshore Wind Turbine System

## Silvio Simani [1,*] and Paolo Castaldi [2]

[1]    Dipartimento di Ingegneria, Università degli Studi di Ferrara, Via Saragat 1E, 44122 Ferrara (FE), Italy
[2]    Dipartimento di Ingegneria dell'Energia Elettrica e dell'Informazione "Guglielmo Marconi"—DEI, Alma Mater Studiorum Università di Bologna, Viale Risorgimento 2, 40136 Bologna (BO), Italy; paolo.castaldi@unibo.it
*    Correspondence: silvio.simani@unife.it; Tel.: +39-0532-97-4844

**Abstract:** Fault diagnosis of wind turbine systems is a challenging process, especially for offshore plants, and the search for solutions motivates the research discussed in this paper. In fact, these systems must have a high degree of reliability and availability to remain functional in specified operating conditions without needing expensive maintenance works. Especially for offshore plants, a clear conflict exists between ensuring a high degree of availability and reducing costly maintenance. Therefore, this paper presents viable fault detection and isolation techniques applied to a wind turbine system. The design of the so-called fault indicator relies on an estimate of the fault using data-driven methods and effective tools for managing partial knowledge of system dynamics, as well as noise and disturbance effects. In particular, the suggested data-driven strategies exploit fuzzy systems and neural networks that are used to determine nonlinear links between measurements and faults. The selected architectures are based on nonlinear autoregressive with exogenous input prototypes, which approximate dynamic relations with arbitrary accuracy. The designed fault diagnosis schemes were verified and validated using a high-fidelity simulator that describes the normal and faulty behavior of a realistic offshore wind turbine plant. Finally, by accounting for the uncertainty and disturbance in the wind turbine simulator, a hardware-in-the-loop test rig was used to assess the proposed methods for robustness and reliability. These aspects are fundamental when the developed fault diagnosis methods are applied to real offshore wind turbines.

**Keywords:** fault diagnosis; analytical redundancy; fuzzy prototypes; neural networks; diagnostic residuals; fault reconstruction; offshore wind turbine simulator

## 1. Introduction

Wind-generated energy is increasingly being used as a power source worldwide, and this has resulted in the need for the enhanced reliability and so-called "sustainability" of wind turbines. Wind turbine systems must continuously generate the required amount of electrical power, depending on the available wind speed, grid demand, and possible malfunctions [1].

Therefore, potential faults affecting the process must be properly detected and managed before causing the deterioration of the nominal working conditions of the plant or becoming critical issues. Wind turbines with large rotors (i.e., of megawatt size) are very expensive systems; they should be highly available and reliable in order to maximize the generated energy (at a reduced cost) and minimize Operation and Maintenance (O&M) services. In fact, most of the cost of the produced energy is from the installation cost of the wind turbine, but unplanned O&M costs could increase it by about 30%, particularly when offshore wind turbines are considered [2].

To this end, many wind turbine systems include conservative technologies that protect against faults, which normally lead to a plant shutdown while awaiting O&M services. Hence, more effective

solutions for managing faults are required to improve wind turbine features, particularly in faulty situations. Such features would prevent critical failures that may affect other wind turbine components, thus avoiding the unplanned replacement of functional parts and increased O&M costs.

It is beneficial to keep maintenance costs as low as possible, decrease downtime, and consequently increase the amount of captured power and improve reliability despite the presence of faults [3]. Fault Detection and Isolation (FDI) techniques are powerful methods for this purpose. The fault information captured by FDI units can be used to optimize maintenance procedures via remote diagnosis [4]. The use of FDI renders the equipment robust to the considered faults and, as a result, maintains the performance of the wind turbine at the desired level, even with the occurrence of faults. Therefore, maintenance requirements and downtime will decrease, and the reliability of power generation will improve. Therefore, the final cost is kept as low as possible [5,6].

FDI designs for wind turbines have been significantly developed over the last decade. Most of the works in this field have been motivated by competitions conducted by kk-electronic a/c and MathWorks from 2009–2016 [4,7]. Accordingly, the number of studies and consequent publications has increased considerably, and the subject is intensively researched worldwide [8]. However, there are only a few available review papers in this field [7,9].

Hardware redundancy involves equipping components, such as sensors and actuators, with physically-identical counterparts to generate so-called residual signatures, which contain information on the possible fault. This approach increases the weight, occupied space, data acquisition complexity, and therefore, the final design cost. These issues are very problematic for offshore wind turbines. In contrast, software redundancy or computer-based FDI techniques have been developed for wind turbines throughout the last decade to overcome the aforementioned problems [1]. A mathematical model of a wind turbine is used to generate redundant signals and, accordingly, residuals.

The most challenging issue, which should be considered in wind turbine FDI schemes, is that wind speed is poorly measured by anemometers due to the spatial/temporal effective wind speed distribution over the blade plane, turbulence, wind shear, and tower shadow effects. Therefore, wind speed is considered an an unknown disturbance, as is the consequent aerodynamic torque. Furthermore, FDI schemes should be robust to the considerable noise present in sensor measurements [4,7].

The most commonly-adopted model-based FDI techniques for wind turbines are the parity relation method and observer design [10]. However, these approaches require accurate mathematical models to simulate the dynamic behaviors of the process under diagnosis [11]. These methods do not require high-resolution signals, so there is no need for data acquisition hardware or installation of additional sensors. However, it is quite challenging to design an effective model that mimics real-world applications. Therefore, data-driven approaches, such as Neural Networks (NN) and fuzzy inference systems, can be used for wind turbine FDI designs. In fact, these artificial intelligence systems provide the best tools to represent the nonlinear and partially-known behavior of wind turbines [12]. The designed prototype is fed with actual/estimated inputs (i.e., those of the wind turbine) to generate redundant outputs. Some other works have proposed the use of this data-driven learning scheme for wind turbine FDI, and it has been considered and applied to different wind turbine components, e.g., gearboxes, generator faults, and pitch faults [10].

As an alternative approach, fault information can be directly extracted/inferred using this method, which relies on the design of an accurate a priori knowledge-based network, e.g., Adaptive Neuro-Fuzzy Inference System (ANFIS) or Fuzzy Inference System (FIS). Accordingly, expert knowledge must be included in the design, whether for numerical rules or fuzzy if/then linguistic rules. One of the advantages of fuzzy logic and fuzzy membership representation is that the uncertain measurement of the wind speed provided by the anemometer can be directly used [8]. Classification methods are also utilized for rotor imbalance/aerodynamic asymmetry fault diagnosis [13].

Therefore, the main contribution of this work is the development of viable and reliable solutions for the fault diagnosis of an offshore wind turbine model. The design of fault-tolerant controllers is not

considered in this paper, but it would likely rely on the same tools considered here. In fact, the fault diagnosis module provides information on the faulty conditions of the system so that the controller activity can compensate. In particular, the FDI task was accomplished here by using fault estimators, which were obtained via these data-driven approaches, as they also offer effective tools for managing limited knowledge of the process dynamics, together with noise and disturbance effects.

The first data-driven solution addressed in this paper relies on fuzzy Takagi–Sugeno models [14], which are derived from a clustering algorithm, followed by an identification procedure [15]. The second solution exploits NN to describe the nonlinear analytical links between measurement and fault signals. The chosen network architecture belongs to the Nonlinear AutoRegressive with eXogenous (NARX) input prototype, which can describe dynamic relationships over time. The training of the neural fault estimators exploits a standard training algorithm that processes the acquired data [16].

The developed fault diagnosis strategies were verified by means of a high-fidelity simulator that describes the normal and faulty behavior of a wind turbine plant. The achieved performances were verified in the presence of uncertainty and disturbance effects, thus validating the reliability and robustness features of the proposed schemes. Their effectiveness, which was further tested using a Hardware-In-the-Loop (HIL) test rig, suggests further investigation of more realistic applications of the proposed schemes.

It is worth noting the rationale underlying the proposal of these tools for the fault diagnosis of wind turbines. When a mathematical description of a plant subject to diagnosis can be included in the FDI design phase, model-based techniques yield the best performances. However, when modeling errors and disturbances are present, the learning phase exploited by the considered data-driven solutions leads to results that are better than those from model-based schemes. In fact, NN and fuzzy models use the learning accumulated from data-driven offline simulations, even if the training stage can be computationally heavy.

This work is organized as follows. Section 2 describes the offshore wind turbine simulator. Section 3 illustrates the fault diagnosis methodologies that rely on fuzzy and NN prototypes. The obtained results are summarized in Section 4, taking into account simulated and real-time conditions. Finally, Section 5 ends the paper by outlining the key achievements of the study and providing suggestions for future research issues.

## 2. Wind Turbine Simulator and Fault Model

The three-bladed horizontal-axis wind turbine model considered in this work follows the principle that wind power activates the wind turbine blades, which leads to the rotation of the low-speed rotor shaft. In order to increase its rotational speed to that which is generally required by the generator, a gearbox with a drivetrain is included in the system. A more detailed description of this benchmark is given in [7], and its schematic diagram is presented in Figure 1.

The wind turbine simulator has two controlled outputs, i.e., the generator rotational speed $\omega_g(t)$ and its generated power $P_g(t)$. The wind turbine model is controlled by means of two actuated inputs, i.e., the generator torque $\tau_g(t)$ and the blade pitch angle $\beta(t)$. The latter signal controls the actuators of the blades, which are implemented by hydraulic drives [7].

Several other measurements are acquired from the wind turbine benchmark: the signal $\omega_r(t)$ represents the rotor speed, and $\tau_r(t)$ is the reference torque. Moreover, the aerodynamic torque signal $\tau_{aero}(t)$ is computed from the wind speed $v(t)$, which is usually available with limited accuracy. In fact, the wind field is not uniform around the wind turbine rotor plane, especially for large rotor systems. Moreover, anemometers measuring this variable are mounted behind the rotor on the nacelle. Therefore, the wind speed measurement $v_w(t)$ is affected by the interference between the blades and the nacelle, as well as the turbulence around the rotor plane. The alteration of the wind speed measurement $v_w(t)$ with respect to its nominal value around the rotor plane represents an uncertainty in the wind turbine model and a disturbance term in the control design [7].

Finally, as sketched in Figure 1, the signals generated by the wind turbine system are assumed to be acquired through the measurement block, whose objective is to simulate the real behavior of the sensors and actuators. Therefore, the measured signals are modeled as the sum of their actual values and white Gaussian process terms. Moreover, the wind turbine simulator includes a baseline controller, represented by standard PID regulators that regulate the generated power on the basis of the actual wind speed, as shown in [4,7].

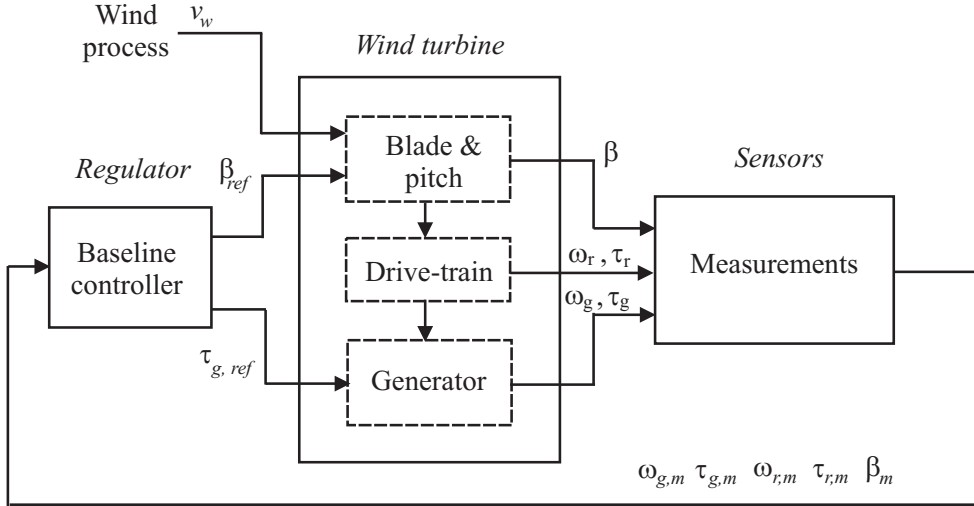

**Figure 1.** Scheme of the offshore wind turbine simulator.

The wind turbine simulator also includes the generation of three different typical fault cases: sensor, actuator, and system faults [4,7]. The sensor faults are generated as additive signals on the affected measurements. As an example, the faulty sensor of the pitch angle $\beta_m$ provides the wrong measurement of the blade orientation, and if not handled, the controller cannot fully track the power reference signal. On the other hand, actuator faults lead to the alteration of the input and output descriptions of the pitch angle and the generator torque models by modifying their dynamics. In this way, a pressure drop in the hydraulic circuit of the pitch actuator and an electronic breakdown in the converter device are simulated, respectively. Finally, a system fault affects the drivetrain of the turbine, which is described as a slow variation in the friction coefficient over time. This can be caused by wear and tear of the mechanical parts over time.

This scenario is summarized in Table 1, which also reports the measured signals that are affected by these nine faults.

**Table 1.** Fault scenario of the wind turbine simulator.

| Fault Case | Fault Type | Affected Measurement |
|:---:|:---:|:---:|
| 1 | Sensor | $\beta_{1,m1}$ |
| 2 | Sensor | $\beta_{2,m2}$ |
| 3 | Sensor | $\beta_{3,m1}$ |
| 4 | Sensor | $\omega_{r,m1}$ |
| 5 | Sensor | $\omega_{r,m2}$ and $\omega_{g,m2}$ |
| 6 | Actuator | Pitch system of Blade #2 |
| 7 | Actuator | Pitch system of Blade #3 |
| 8 | Actuator | $\tau_{g,m}$ |
| 9 | System | Drivetrain |

The overall model of the wind turbine process is represented as a nonlinear continuous-time function $\mathbf{f}_{wt}$ that describes the evolution of the turbine state vector $\mathbf{x}_{wt}$ excited by the input vector $\mathbf{u}$:

$$\begin{cases} \dot{\mathbf{x}}_{wt}(t) & = & \mathbf{f}_{wt}\left(\mathbf{x}_{wt}, \mathbf{u}(t)\right) \\ \mathbf{y}(t) & = & \mathbf{x}_{wt}(t) \end{cases} \tag{1}$$

where, in this case, the state of the system is considered equal to the outputs of the wind turbine system, i.e., the rotor speed, the generator speed, and the generated power:

$$\mathbf{x}_{wt}(t) = \mathbf{y}(t) = \left[\omega_{g,m1}, \omega_{g,m2}, \omega_{r,m1}, \omega_{r,m2}, P_{g,m}\right]$$

On the other hand, the input vector,

$$\mathbf{u}(t) = \left[\beta_{1,m1}, \beta_{1,m2}, \beta_{2,m1}, \beta_{2,m2}, \beta_{3,m1}, \beta_{3,m2}, \tau_{g,m}\right]$$

consists of the measurements of the three pitch angles from the three redundant sensors, as well as the measured torque. These signals are sampled with a sample time $T$ in order to acquire a total of $N$ measurements $\mathbf{u}(k)$, $\mathbf{y}(k)$ with $k = 1, \ldots, N$, in order to implement the data-driven fault diagnosis solutions proposed in this paper.

It is worth noting that, as highlighted in Section 3, the effect of the faults considered in Table 1 is assumed to be generated by equivalent signals added to the input and output measurements. This approach was formerly proposed by the authors of [17]. Moreover, this assumption is also known as Errors-In-Variables (EIV) modeling, which is exploited in the dynamic system identification framework [18].

## 3. Fault Diagnosis Techniques: Fuzzy Systems and Neural Networks

In order to solve the fault diagnosis problem, this work assumes that the wind turbine system is affected by equivalent additive faults on the input and output measurements, as well as measurement errors, as described by the relations in Equation (2):

$$\begin{cases} \mathbf{u}(k) & = & \mathbf{u}^*(k) + \tilde{\mathbf{u}}(k) + \mathbf{f}_u(k) \\ \mathbf{y}(k) & = & \mathbf{y}^*(k) + \tilde{\mathbf{y}}(k) + \mathbf{f}_y(k) \end{cases} \tag{2}$$

where $\mathbf{u}^*(k)$ and $\mathbf{y}^*(k)$ represent the actual process variables; $\mathbf{u}(k)$ and $\mathbf{y}(k)$ are the measurements acquired by the sensors; and $\tilde{\mathbf{u}}(k)$ and $\tilde{\mathbf{y}}(k)$ describe the measurement errors. Note that, according to the relations in Equation (2), it is assumed that the fault signals $\mathbf{f}_u(k)$ and $\mathbf{f}_y(k)$ have equivalent additive effects. These functions are different from zero only in the presence of faults. In general, the vector $\mathbf{u}(k)$ has $r$ components, i.e., the number of process inputs, while $\mathbf{y}(k)$ has $m$ elements, i.e., the number of process outputs.

This work suggests exploiting fuzzy system and NN structures in order to provide an online estimation $\hat{\mathbf{f}}(k)$ of the fault signals $\mathbf{f}_u(k)$ and $\mathbf{f}_y(k)$. Hence, as shown in Figure 2, the diagnostic residuals $\mathbf{r}(k)$ are equal to the estimated fault signals, $\hat{\mathbf{f}}(k)$, as in Equation (3):

$$\mathbf{r}(k) = \hat{\mathbf{f}}(k) \tag{3}$$

The variable $\hat{\mathbf{f}}(k)$ is the fault vector, i.e., $\hat{\mathbf{f}}(k) = \left\{\hat{f}_1(k), \ldots, \hat{f}_{r+m}(k)\right\}$. Therefore, the general fault estimate $\hat{f}_i(k)$ is equal to the $i^{textth}$ component of the fault vectors $\mathbf{f}_u(k)$ or $\mathbf{f}_y(k)$ in Equation (2), with $i = 1, \ldots, r + m$. This residual generation scheme is represented in Figure 2.

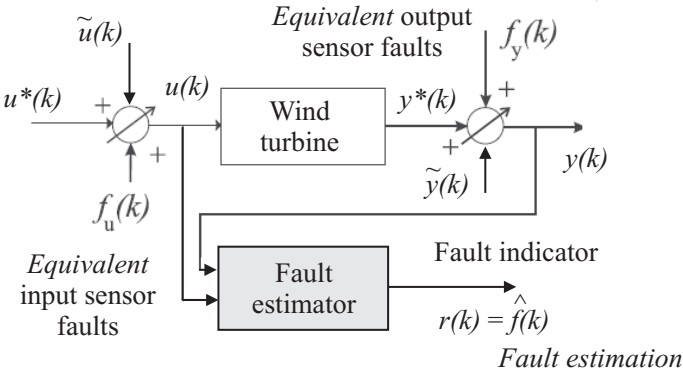

**Figure 2.** The residual generation scheme.

Figure 2 shows that, in general, the residual generators are fed by the input and output measurements $\mathbf{u}(k)$ and $\mathbf{y}(k)$. The occurrence of the $i^{textth}$ fault can be simply detected using the threshold logic of Equation (4) applied to the $i^{textth}$ residual $r_i(k)$ [11]:

$$\begin{cases} \bar{r}_i - \delta\sigma_{r_i} \leq r_i \leq \bar{r}_i + \delta\sigma_i & \text{fault–free case} \\ r_i < \bar{r}_i - \delta\sigma_{r_i} \text{ or } r_i > \bar{r}_i + \delta\sigma_{r_i} & \text{faulty case} \end{cases} \tag{4}$$

with $r_i(k)$ representing the $i^{textth}$ component of the vector $\mathbf{r}(k)$. Its mean $\bar{r}_i$ and variance $\sigma_{r_i}^2$ values are computed in a fault-free condition from $N$ samples according to the relations in Equation (5):

$$\begin{cases} \bar{r}_i &= \frac{1}{N} \sum_{k=1}^{N} r_i(k) \\ \sigma_{r_i}^2 &= \frac{1}{N} \sum_{k=1}^{N} (r_i(k) - \bar{r}_i)^2 \end{cases} \tag{5}$$

Note that the parameter $\delta$ represents a variable that has to be properly tuned in order to separate the fault-free from the faulty conditions effectively, as shown in Section 4. Once the fault detection phase is complete, the fault isolation task is directly obtained by means of the bank of estimators depicted in Figure 3.

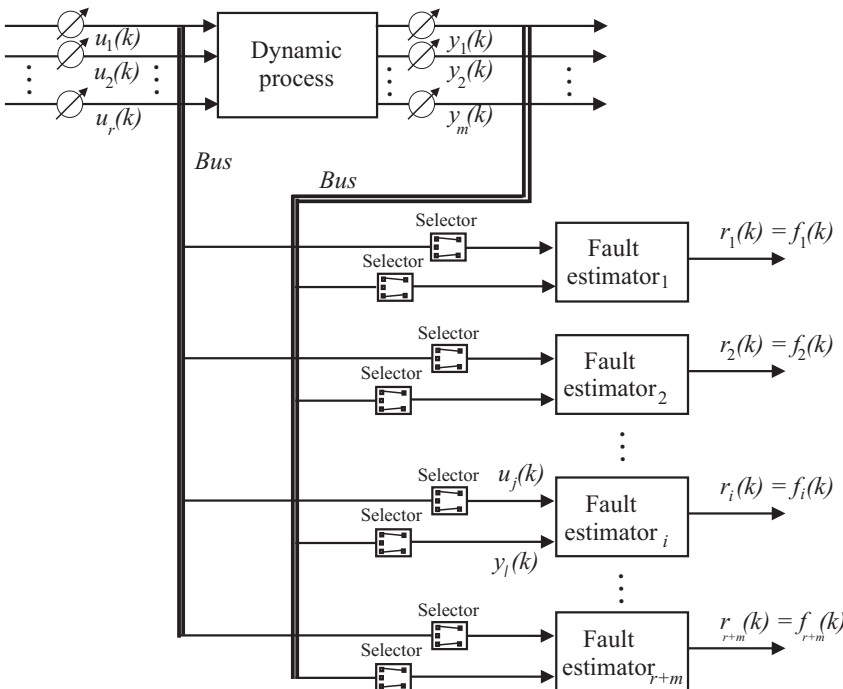

**Figure 3.** The estimator scheme for the reconstruction of the equivalent input or output faults $f_i(t)$.

According to the scheme depicted in Figure 3, the number of estimators in the bank is equal to the number of faults that have to be diagnosed, i.e., $r + m$. In general, the $i^{textth}$ estimator is driven by the input and output signals $\mathbf{u}(k)$ and $\mathbf{y}(k)$. However, its inputs $u_j(k)$ and output $y_l(k)$ are selected in order to be selectively sensitive to the particular fault $f_i(t)$. To this end, the design of these fault estimators is enhanced by the fault sensitivity analysis procedure reported in Section 3.

The first method proposed in this paper for designing fault estimators relies on Takagi–Sugeno (TS) models [19]. This approach was formerly addressed in [14] for the approximation of nonlinear Multiple-Input Single-Output (MISO) dynamic systems with arbitrary accuracy. The general fault estimator $\hat{f}$ has the form of Equation (6):

$$\hat{f} = \frac{\sum_{i=1}^{n_C} \lambda_i(\mathbf{x}) \left( \mathbf{a}_i^T \mathbf{x} + b_i \right)}{\sum_{i=1}^{n_C} \lambda_i(\mathbf{x})} \tag{6}$$

The TS fuzzy model results are described as discrete-time linear AutoRegressive models with eXogenous input (ARX) of order $o$, in which the regressor vector has the form of Equation (7):

$$\mathbf{x}(k) = \left[ \ldots, y_l(k-1), \ldots, y_l(k-o), \ldots u_j(k), \ldots, u_j(k-o), \ldots \right]^T \tag{7}$$

where $u_l(\cdot)$ and $y_j(\cdot)$ are the components of the actual system input and output vectors $\mathbf{u}(k)$ and $\mathbf{y}(k)$ that are selected using the fault sensitivity analysis proposed in Section 3. The variable $k$ represents the time step, with $k = 1, 2, \ldots, N$. The parameters of the TS fuzzy model in Equation (6) are collected into the vector:

$$\mathbf{a}_i = \left[ \alpha_1^{(i)}, \ldots, \alpha_o^{(i)}, \delta_1^{(i)}, \ldots, \delta_o^{(i)} \right]^T \tag{8}$$

where the $\alpha_j^{(i)}$ coefficients refer to the output samples, while the $\delta_j^{(i)}$ coefficients are associated with the input ones.

This work proposes to solve the derivation of the TS models as a system identification problems from the noisy data of Equation (2). In particular, the design of the bank of fault estimators in Figure 3 requires the estimation of the consequent parameters $\mathbf{a}_i$ and $b_i$ of Equation (8).

Note that the design method proposed in this work exploits the direct identification of the TS fuzzy models of Equation (6). In particular, the fuzzy model structure, i.e., the number of rules $n_C$, the antecedents, and the fuzzy membership functions $\lambda_i(\mathbf{x})$ in Equation (6), are derived by means of the Fuzzy Modeling and Identification (FMID) toolbox implemented in the MATLAB environment [14]. Moreover, the computation of the TS model parameters in Equation (8) was solved by the authors in [20] as an EIV estimation problem, as highlighted by the relations in Equation (2). On the other hand, the FMID toolbox uses the Gustafson–Kessel (GK) clustering method [14] to perform a partition of input–output data into a proper number $n_C$ of regions (clusters), where the $i^{textth}$ model of Equation (6) is valid. This model is thus obtained after the selection of the model order $o$ and the number of clusters $n_C$. The FMID toolbox also determines the antecedent degrees of fulfillment $\lambda_i(\mathbf{x})$ in Equation (6), which are derived with a curve fitting method [14].

This paper proposes a different data-driven approach that is based on NN, which is exploited to implement the scheme shown in Figure 3. According to this scheme, a bank of NN is used to reconstruct the faults affecting the system under diagnosis using a proper set of input and output measurements. The structure proposed in this work consists of a feedforward multilayer perceptron NN with three layers [21]. Moreover, this study suggests the use of a quasi-static NN, as it represents a suitable tool to predict dynamic relationships between the input–output measurements and the considered fault function $f_i(k)$ with arbitrary accuracy [21].

Therefore, the $i^{textth}$ neural fault estimator in Figure 3 is described by the relation in Equation (9):

$$\hat{f}_i(k) = F \left( \ldots, u_j(k), \ldots, u_j(k-d_u), \ldots y_l(k-1), \ldots, y_l(k-d_y), \ldots \right) \tag{9}$$

where $u_j(\cdot)$ and $y_l(\cdot)$ are the general $j^{textth}$ and $l^{textth}$ components of the measured inputs and outputs **u** and **y**, respectively, that are selected via the fault sensitivity analysis tool. $d_u$ and $d_y$ represent the number of delays of the input and the output samples. $F(\cdot)$ is the function realized by the static NN, which depends on the number of neurons and their weights.

The NN exploited in this study uses sigmoidal activation functions for the neurons in both the input and the hidden layers, while a linear one is used in the output layer. The number of neurons and delays ($d_u$ and $d_y$) is selected to obtain suitable fault estimation errors after the NN training from the data acquired from the system under diagnosis. In particular, the NN training is performed by generating a proper number of data, $N$, which are partitioned into the training, validation, and test sets, as required by the Levenberg–Marquardt back-propagation algorithm [21].

*Fault Sensitivity Analysis*

The design of the fault diagnosis schemes proposed in this paper and represented in Figure 3 is enhanced by the tool presented here. It consists of a fault sensitivity analysis that is performed on the measurements acquired from the wind turbine simulator. The procedure aims to define the most sensitive measurements $u_j(k)$ and $y_l(k)$ with respect to the general fault $f_i(k)$ considered in Section 2.

According to the assumption of Equation (2), the considered fault signals $f_i(k)$ have been injected into the wind turbine simulator, and only single faults may occur. Then, the Relative-Mean-Squared Errors (RMSEs) between the fault-free and faulty signals acquired from the simulator are computed. In this way, the most sensitive signals $u_j(k)$ and $y_l(k)$ are selected for each fault $i$. The achieved results are summarized in Table 2.

**Table 2.** The most sensitive measurements $u_j(k)$ and $y_l(k)$ and their RMSE values with respect to the fault $f_i(k)$.

| Fault $f_i$ | 1 | 2 | 3 | 4 | 5 | 6 | 7 | 8 | 9 |
|---|---|---|---|---|---|---|---|---|---|
| **Measurements $u_j$, $y_l$** | $\beta_{1,m1}$ | $\beta_{2,m2}$ | $\beta_{3,m1}$ | $\omega_{r,m1}$ | $\omega_{r,m1}$ | $\beta_{2,m1}$ | $\beta_{3,m2}$ | $\tau_{g,m}$ | $\omega_{g,m1}$ |
| **RMSE** | 11.29 | 0.98 | 2.48 | 1.44 | 1.45 | 0.80 | 0.73 | 0.84 | 0.77 |

In particular, the fault sensitivity analysis follows the selection algorithm, which relies on the normalized sensitivity function $N_x$ of Equation (10),

$$N_x = \frac{S_x}{S_x^*} \tag{10}$$

with:

$$S_x = \frac{\left\| x_f(k) - x_n(k) \right\|_2}{\| x_n(k) \|_2} \tag{11}$$

and:

$$S_x^* = \max \frac{\left\| x_f(k) - x_n(k) \right\|_2}{\| x_n(k) \|_2} \tag{12}$$

In fact, $N_x$ represents the effect of the considered fault case with respect to the measured signal $x(k)$, with $k = 1, 2, \ldots, N$. The subscripts "$f$" and "$n$" indicate the faulty and the fault-free cases, respectively. Therefore, the measurement that is most affected by the considered fault is the value of $N_x$, which, in this case, is equal to one. Otherwise, smaller values of $N_x$ indicate that $x(k)$ is not affected by that fault.

The complete results of the fault sensitivity analysis are summarized in Table 3.

**Table 3.** The most sensitive measurements with respect to the considered fault scenario.

| Fault Case $f_i$ | Most Sensitive Inputs $u_j$ | Most Sensitive Outputs $y_l$ |
|---|---|---|
| 1 | $\beta_{1,m1}, \beta_{1,m2}$ | $\omega_{g,m2}$ |
| 2 | $\beta_{1,m2}, \beta_{2,m2}$ | $\omega_{g,m2}$ |
| 3 | $\beta_{1,m2}, \beta_{3,m1}$ | $\omega_{g,m2}$ |
| 4 | $\beta_{1,m2}$ | $\omega_{g,m2}, \omega_{r,m1}$ |
| 5 | $\beta_{1,m2}$ | $\omega_{g,m2}, \omega_{r,m2}$ |
| 6 | $\beta_{1,m2}, \beta_{2,m1}$ | $\omega_{g,m2}$ |
| 7 | $\beta_{1,m2}, \beta_{3,m2}$ | $\omega_{g,m2}$ |
| 8 | $\beta_{1,m2}, \tau_{g,m}$ | $\omega_{g,m2}$ |
| 9 | $\beta_{1,m2}$ | $\omega_{g,m1}, \omega_{g,m2}$ |

This method represents a key feature of the proposed approach to fault diagnosis. In fact, the fault estimators of the bank of Figure 3 are designed by exploiting a reduced number of input signals $u_j(k)$ and $y_l(k)$. It also leads to a noteworthy simplification of the complexity and the computational cost of the identification and training phases of the fuzzy and NN models, respectively.

Note finally that the fault sensitivity analysis was performed by considering one fault at a time. The case of multiple faults was not considered here, as the wind turbine benchmark simulates the occurrence of single faults only, as described in [4,7]. However, the case of multiple faults occurring at the same time could be considered, even if a different fault sensitivity analysis has to be executed.

## 4. Performance and Robustness Analysis

This section addresses the evaluation of the performances of the fault diagnosis strategies described in Section 3. In particular, Section 4.1 considers the simulations from the wind turbine benchmark of Section 2. On the other hand, in order to assess the effectiveness of the considered solutions in a more realistic framework, Section 4.2 considers HIL experiments obtained by means of an industrial computer interacting with onboard electronics.

### 4.1. Simulation Results

With reference to the wind turbine benchmark in Section 2, all simulations were driven by the same wind sequence $v_w(t)$. It represents a real measurement of wind speed, from 5–20 m/s, with a few spikes at 25 m/s. Moreover, the rated power of the wind turbine is $P_r = 4.8$ MW, and the nominal generator speed is $\omega_{nom} = 162.5$ rad/s [7]. The simulations lasted for 4400 s with single fault occurrences. The measurements were acquired with a sampling frequency of 100 Hz, so $N = 440{,}000$ samples were generated for each run. Table 4 summarizes the wind turbine fault modes, as described in Section 2.

**Table 4.** Fault modes of the wind turbine simulator.

| Fault Case | Fault Type | Fault Shape | Occurrence (s) |
|---|---|---|---|
| 1 | actuator | step | 2000–2100 |
| 2 | actuator | step | 2300–2400 |
| 3 | actuator | step | 2600–2700 |
| 4 | actuator | step | 1500–1600 |
| 5 | actuator | step | 1000–1100 |
| 6 | sensor | step | 2900–3000 |
| 7 | sensor | trapezoidal | 3500–3600 |
| 8 | sensor | step | 3800–3900 |
| 9 | sensor | step | 4100–4300 |

Note that Fault Case 7 reported in Table 4 is modeled with a trapezoidal function, which is directly added to the corresponding output measurement according to the model in Equation (2). On the other hand, Fault Case 9 is generated as a step change of the parameters of the transfer function describing the drivetrain model. However, the effect of this fault on the output measurements is different from a step function. More details regarding the wind turbine fault scenario can be found in [4,7].

As an example, in order to show different fault effects on process measurements, Figure 4 compares the results of the fault sensitivity test in terms of fault-free and faulty signals. In particular, Faults 1, 2, 3, and 8 are considered.

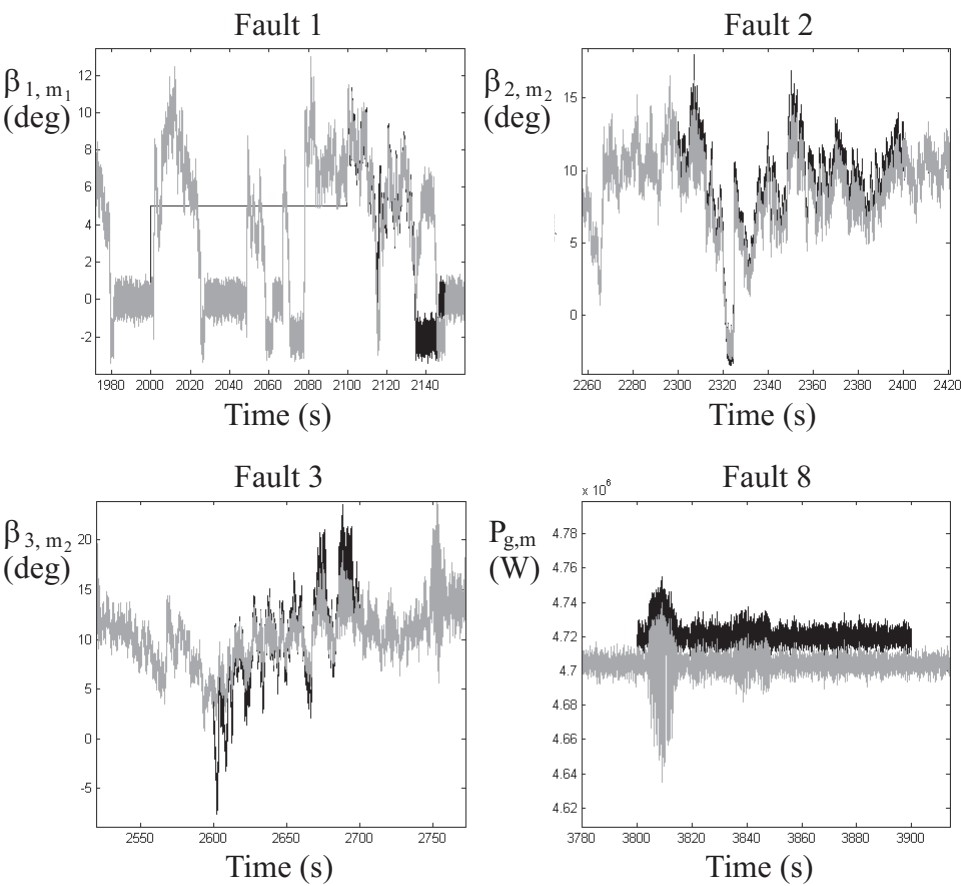

**Figure 4.** The fault-free (gray line) signals with respect to the faulty ones (black line).

When the FMID tool was applied to the data of the wind turbine simulator, $n_C = 4$ clusters and $o = 3$ delays to input and output regressors of the TS fuzzy models were determined. This tool also provided the membership function points, which were fitted through Gaussian membership functions [14]. The optimal values of $n_C$ and $o$ were determined in order to minimize the fuzzy model estimation errors. After data clustering, the regressands $\alpha_j^{(i)}$ and $\delta_j^{(i)}$ in Equation (8) were identified. The TS models in Equation (6) were thus implemented, and nine fault estimators were organized with the bank structure of Figure 3. Note that, according to Table 3, each fuzzy fault estimator in Equation (6) has three inputs. Therefore, each TS fuzzy model has a number of parameters equal to $(3 + 1) \times n = 12$.

The capabilities of the TS fuzzy estimators were assessed in terms of Root-Mean-Squared Error (RMSE), which is computed as the difference between the predicted $\hat{f}_i(k)$ and the actual fault $f_i(k)$, with $i = 1, \ldots, 9$. Table 5 summarizes the achieved performance of the nine TS fuzzy fault estimators.

**Table 5.** Fault estimator performance in terms of RMSE.

| Fault Estimator $i$ | 1 | 2 | 3 | 4 | 5 | 6 | 7 | 8 | 9 |
|---|---|---|---|---|---|---|---|---|---|
| RMSE | 0.016 | 0.023 | 0.021 | 0.020 | 0.019 | 0.021 | 0.017 | 0.021 | 0.019 |

In order to perform the fault detection task, the diagnostic residuals $r_i(k) = \hat{f}_i(k)$ were compared according to the threshold logic of Equation (4). The parameter $\delta$ has to be selected in order to optimize the fault diagnosis performance: for example, in terms of missed faults and false alarm rates [22]. Table 6 summarizes the values of this parameter for each fault estimator $i$.

**Table 6.** Optimal value of the parameter $\delta$.

| Residual $r_i(k)$ | 1 | 2 | 3 | 4 | 5 | 6 | 7 | 8 | 9 |
|---|---|---|---|---|---|---|---|---|---|
| $\delta$ | 3.8 | 4.3 | 4.2 | 4.5 | 3.7 | 4.4 | 4.3 | 3.5 | 3.9 |

In the following, the simulation results are reported, particularly for Fault Cases 1, 4, 8, and 9. The estimated faults $\hat{f}_i$ depicted in Figure 5 demonstrate that the fault detection task was achieved, as they exceeded the threshold levels only when the corresponding fault was active, as reported in Table 4.

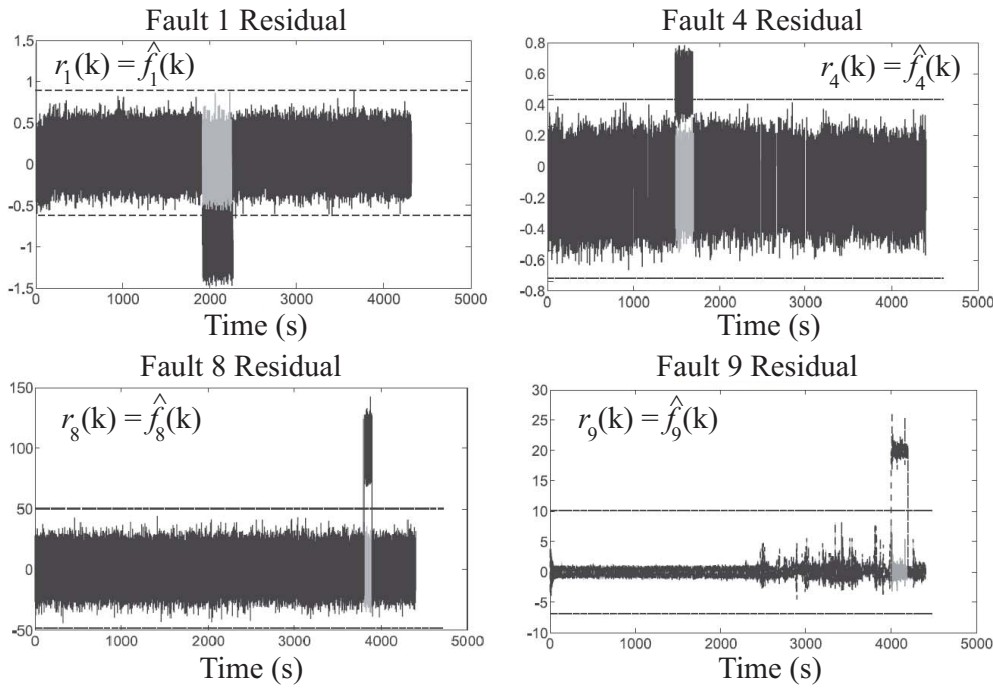

**Figure 5.** The estimated faults $\hat{f}_i$ for Cases 1, 4, 8, and 9.

Figure 5 depicts the reconstructed fault functions $\hat{f}_i(k)$ generated by the fuzzy estimators in faulty conditions (black continuous line) with respect to the fault-free residuals (gray line). The fixed thresholds of Equation (4) are depicted by dotted lines. It is worth noting that in fault-free conditions, the estimated fault functions $\hat{f}_i(k)$ are not zero due to the model–reality mismatch and the measurement error. The results also highlight the robustness and reliability characteristics of the developed fault diagnosis technique, which relies on the proposed fuzzy tool.

For the fuzzy systems, nine NARX NN models were designed according to the scheme in Figure 3. The NN structure selected in this study consisted of 3 layers, with 3 neurons in the input layer, 8 in the hidden one, and 1 neuron in the output layer. Furthermore, in this case, a trial and error procedure was used to determine the optimal number of delays $d_u$ and $d_y$, as well as the number of neurons, that led

to the minimization of the fault estimation error. In particular, $d_u = d_y = 4$ delays were selected in the relation of Equation (9). According to Table 3 and Figure 3, the NN models have three inputs.

The prediction capabilities of the neural fault estimators are summarized in Table 7, which reports the values of the RMSEs obtained by comparing the estimated faults with the simulated ones.

**Table 7.** NN performance in terms of RMSE.

| Fault Estimator $i$ | 1 | 2 | 3 | 4 | 5 | 6 | 7 | 8 | 9 |
|---|---|---|---|---|---|---|---|---|---|
| RMSE | 0.009 | 0.009 | 0.009 | 0.012 | 0.011 | 0.011 | 0.009 | 0.009 | 0.014 |

Furthermore, in this case, the fault detection task was achieved by comparing the residuals $r_i = \hat{f}_i(k)$ from the neural fault estimators with the optimized thresholds of Equation (4). The values of the parameter $\delta$ are reported in Table 8.

**Table 8.** $\delta$ values for the threshold logic.

| Residual $r_i(k)$ | 1 | 2 | 3 | 4 | 5 | 6 | 7 | 8 | 9 |
|---|---|---|---|---|---|---|---|---|---|
| $\delta$ | | 4.2 | 4.9 | 4.7 | 5.1 | 4.2 | 4.6 | 4.8 | 4.1 | 4.3 |

As an example, with reference to Fault Cases 1, 2, 3, and 4, Figure 6 depicts the residuals $\hat{f}_i(k)$ generated in faulty conditions by the NN estimators (continuous line) compared with the fixed thresholds (dashed line).

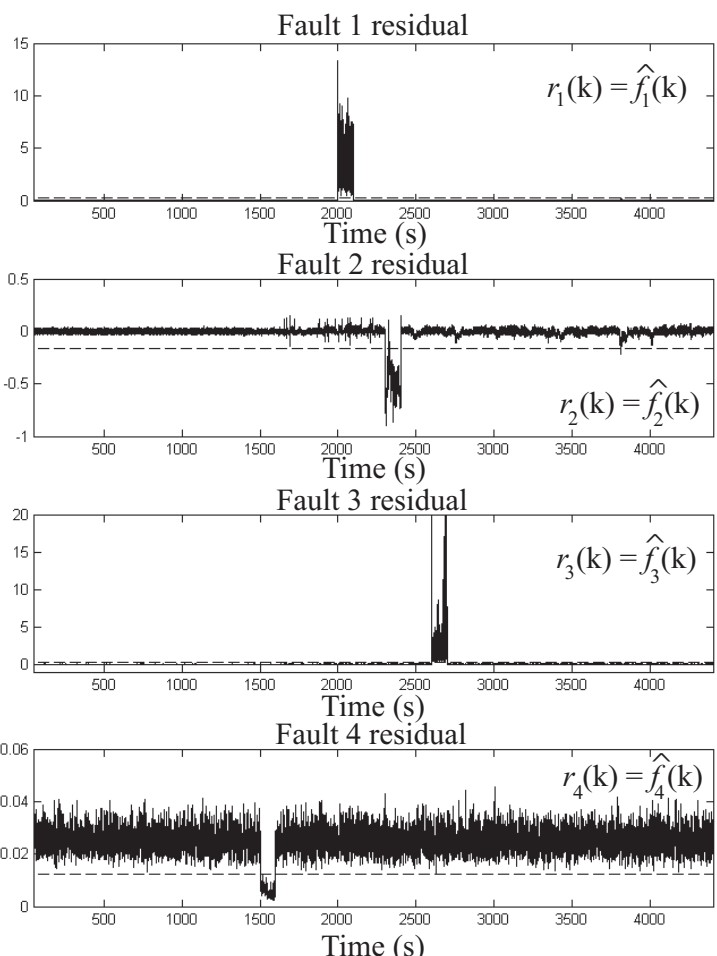

**Figure 6.** Estimated signals (continuous line) $\hat{f}_i(k)$ and fixed thresholds (dashed line) for Faults 1, 2, 3, and 4.

Furthermore, in this case, the achieved results show the effectiveness of the proposed fault diagnosis solutions with respect to disturbance and uncertainty effects simulated by the wind turbine benchmark, thus highlighting their potential application to real wind turbine systems.

### 4.2. Hardware-in-the-Loop Experiments

The HIL test rig was implemented in order to validate the proposed fault diagnosis schemes in real-time conditions. This tool was formerly considered in [23], but for fault-tolerant control design purposes.

The experimental setup in Figure 7 consists of three interconnected components:

- Simulator: The offshore wind turbine system summarized in Section 2 was implemented in the LabVIEW® environment. This software tool runs on an industrial CPU, which allows real-time monitoring of the simulated system parameters.
- Onboard electronics: The fault diagnosis schemes were implemented in the AWC 500 system, which features standard wind turbine specifications. This element acquires the signals from the wind turbine simulator and processes the fault diagnosis solutions proposed in this study.
- Interface circuits: These facilitate communication between the simulator and the onboard electronics.

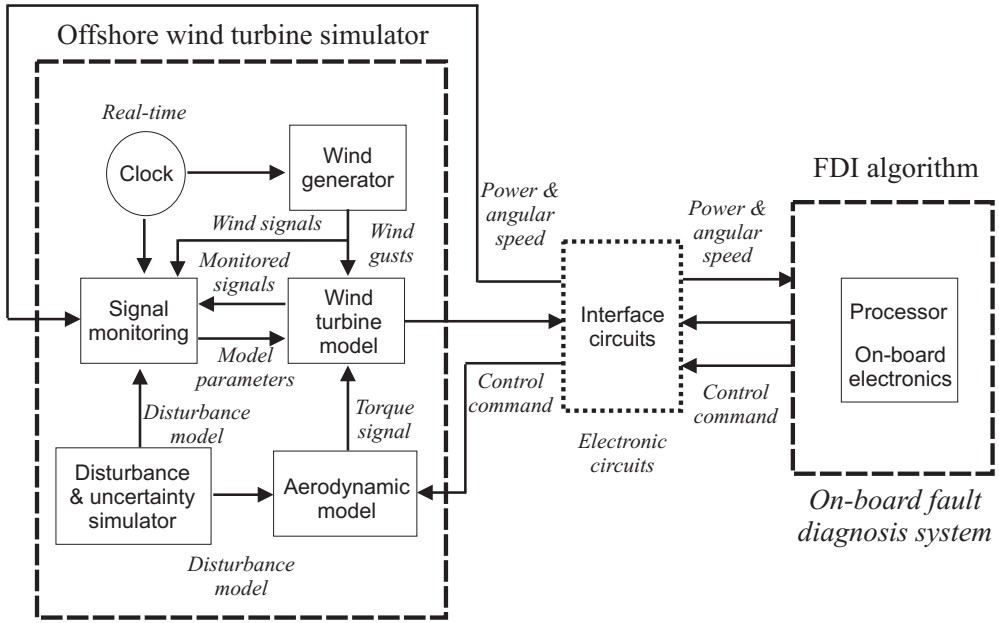

**Figure 7.** The block diagram of the HIL test rig. FDI, Fault Detection and Isolation.

The achieved performances were evaluated on the basis of the following computed indices, which were formerly proposed in [24]:

- False Alarm Rate (FAR): the ratio between the number of wrongly-detected faults and the number of simulated faults;
- Missed Fault Rate (MFR): the ratio between the total number of missed faults and the number of simulated faults;
- True FDI Rate (TFR): the ratio between the number of correctly-detected faults and the number of simulated faults;
- Mean FDI Delay (MFD): the average time delay between fault occurrence and fault detection.

A total of 1000 experiments were performed in order to compute these indices, as the efficacy of the developed fault diagnosis techniques depends on the model–reality mismatch and the actual measurements errors.

Table 9 summarizes the results obtained by implementing fuzzy estimators using the real-time HIL setup.

**Table 9.** Performance indices with fuzzy fault estimators. MFR, Missed Fault Rate; TFR, True FDI Rate; MFD, Mean FDI Delay.

| Estimated Fault $\hat{f}_i(k)$ | FAR | MFR | TFR | MFD |
|:---:|:---:|:---:|:---:|:---:|
| 1 | 0.005 | 0.005 | 0.995 | 0.077 |
| 2 | 0.004 | 0.004 | 0.996 | 0.490 |
| 3 | 0.004 | 0.004 | 0.996 | 0.080 |
| 4 | 0.005 | 0.005 | 0.995 | 0.070 |
| 5 | 0.003 | 0.004 | 0.997 | 0.060 |
| 6 | 0.004 | 0.005 | 0.996 | 0.760 |
| 7 | 0.005 | 0.004 | 0.995 | 0.640 |
| 8 | 0.005 | 0.004 | 0.995 | 0.060 |
| 9 | 0.004 | 0.005 | 0.996 | 0.180 |

On the other hand, Table 10 reports the values achieved with the NN fault estimators implemented using the same real-time HIL setup.

**Table 10.** Performance indices with NN fault estimators.

| Estimated Fault $\hat{f}_i(k)$ | FAR | MFR | TFR | MFD |
|:---:|:---:|:---:|:---:|:---:|
| 1 | 0.007 | 0.006 | 0.899 | 0.014 |
| 2 | 0.234 | 0.005 | 0.867 | 0.516 |
| 3 | 0.004 | 0.004 | 0.914 | 0.080 |
| 4 | 0.005 | 0.005 | 0.922 | 0.070 |
| 5 | 0.006 | 0.007 | 0.905 | 0.097 |
| 6 | 0.005 | 0.006 | 0.989 | 0.871 |
| 7 | 0.701 | 0.007 | 0.981 | 6.987 |
| 8 | 0.498 | 0.008 | 0.987 | 0.289 |
| 9 | 0.197 | 0.176 | 0.798 | 0.399 |

Some further remarks can be made here. When an accurate mathematical description of the system under diagnosis can be included in the design phase, model-based fault diagnosis techniques may yield the best performances. However, when modeling errors and uncertainty are present, the optimization and learning exploited by the proposed data-driven solutions lead to very accurate results. In fact, the TS fuzzy models led to interesting fault diagnosis capabilities, as they used the adaptation accumulated from offline simulations. On the other hand, the NN structures use the training stage, which can be computationally heavy. It can thus be concluded that the proposed data-driven approaches seem to represent powerful techniques that are able to cope with uncertainty and disturbances, as well as variable working conditions.

Finally, the results reported here confirm the effectiveness of the developed fault diagnosis schemes when applied to a real-time test rig. Moreover, the robustness features of the proposed solutions support the viability of applying the proposed fault diagnosis techniques to real offshore wind turbine systems.

## 5. Conclusions

This paper presents the development and analysis of practical tools for performing fault diagnosis of a wind turbine system. The design of this indicator relies on the direct estimate of the fault itself and uses two data-driven schemes. These are proposed by the authors to be viable tools for coping

with poor knowledge of the process dynamics in the presence of noise and disturbance effects. These data-driven schemes are based on fuzzy and neural network structures used to derive the nonlinear dynamic link between the input–output measurements and the considered fault signals. The selected prototypes belong to nonlinear autoregressive with exogenous input architectures, as they can describe any nonlinear dynamic relationship with an arbitrary degree of accuracy. The fault diagnosis strategies were tested via a high-fidelity simulator describing the normal and faulty behaviors of an offshore wind turbine plant. The achieved performances, in terms of reliability and robustness, were thus verified by considering the presence of uncertainty and disturbance effects simulated by the wind turbine benchmark. In order to assess the considered fault diagnosis solutions in a more realistic framework, hardware-in-the-loop experiments were also analyzed by means of an industrial computer interacting with onboard electronics. The achieved results highlight that data-driven approaches, such as fuzzy systems and neural networks, are able to lead to robust and reliable solutions, even if optimization and adaptation procedures are required. Further works will consider the application of these fault diagnosis schemes to real plants.

Sample availability: The software simulation codes for the proposed fault diagnosis strategies and the proposed results are available from the authors in the Matlab and Simulink environments.

**Author Contributions:** S.S. conceived of and designed the simulations; moreover, he analyzed the methodologies and the achieved results; together with P.C., he also wrote the paper.

**Funding:** This research received no external funding.

**Acknowledgments:** The research work has been supported by the FAR2018local fund from the University of Ferrara. The costs to publish in open access have been covered by the FIR2018local fund from the University of Ferrara.

**Conflicts of Interest:** The authors declare no conflicts of interest.

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
