# Peer review of "Intelligent Fault Diagnosis Techniques Applied to an Offshore Wind Turbine System"

_applsci, doi:10.3390/app9040783_

Round 1

Reviewer 1 Report

The issue this manuscript trying to address is interesting and important.

One main contribution of the paper to apply Fuzzy and NN diagnostic algorithm on an offshore wind turbine model.

Please enrich the literature review for better follow up and to clarify the need of the study.

Overall, the presentation is OK and is recommended for a through language polishing and  another round of review after the comments detailed below are addressed.

Abstract:

In general it should be edited to make the contribution of the paper more clear.

a)     L2; “offshore installation” it is two sided! You mean the installation operation of an offshore wind turbine or offshore wind turbine itself. Please make it clear.

b)     L1-2; Language: “The fault diagnosis of wind turbine systems represent a challenging issue, *, thus justifying the research topics developed in this work.

c)     L3; Typo: it represents

d)     L6; Language: “practical analytical knowledge” parallel structure adj+and+adj or adv+adj

Practical and analytical knowledge.

1-     Introduction

The introduction is really poor in terms of presented adequate and relevant references which lead to the niche of the paper. For example,

a)     L36: Language and clarity. Like: “this will lead to prevent critical failures” change to “this would prevent….”

b)      L43,”[2,3]”: maybe better to cite review papers if you want to point general research in the field.

c)     L44, lack of reference: “at a wind turbine level”

d)     L48-50, redundant and doesn’t convey useful information!

e)     There are some language issues in the other paragraphs and the literature review is not systematically presented to direct the reader to the need of what has been done in the paper with proper referencing in each step. Please revise the section completely.

2-     Offshore wind turbine simulator

The main issue in this section is that it refers to 3 different fault cases: sensor, actuator and system faults. However for the modeling only additive faults have been considered. While, the nature of change in the system dynamics is multiplicative faults so how to comment on that?

 3. Intelligent Fault Diagnosis Techniques
Generally the language and typos should be revised. Like

a)     L127: Typo: “relying on on fuzzy system”.

b)     L217, Language:“regards”

c)     How to derive the fuzzy rules? Any algorithm to obtained the optimal ones?

and didn’t comment on the choice of the fuzzy topology.

d)     Neural Network: why this architect of the network? Please comment of the training data and approach, testing data and number of neurons.

e)     Table3: sensitivity fault analysis has been done by considering one fault at a time, yes? However, in reality there might be multiple faults at the same time with might have effect on the nature of the system and change the sensitivity, how to comment on this?

f)      Table3: I am wondering due to symmetry, why, in Faults 1,2,3  is repeating and not for blade 2 or 3. And for the output why not the  and not  if they are redundant sensors.

The same issue for faults 6,7.

 4. Results and Discussion
Again language problem.

a) L303-305: Language, vague!

b) L293: please comment on the WT, rated power, wind speed …

c) Table4: why fault case 7 is considered as trapezoidal while the fault case 9 is more of a nature of gradual change due to for example wear in drive train?

d) Table 9: the presented results like FAR and MFR is really poor how to comment on it?

Author Response

The paper has been revised according to the comments received by the two anonymous referees. All changes have been highlighted in yellow in the revised version of the paper and the responses to reviewers’ comments are reported in the following.

The authors would like to thank the Managing Editor and the anonymous referee for her/his comments and suggestions, which were taken into account for improving the revised version of the manuscript. In the following, the anonymous referee’s comments and remarks are summarised (in black colour text) and followed by the authors’ answers, highlighted in blue colour. Moreover, the modified parts of the manuscript have been highlighted in yellow.

Anonymous Referee #1 Comments

The issue this manuscript trying to address is interesting and important.

The authors would like to thank the anonymous referee for this remark.

One main contribution of the paper to apply Fuzzy and NN diagnostic algorithm on an offshore wind turbine model.

Please enrich the literature review for better follow up and to clarify the need of the study.

The authors would like to thank the anonymous referee for this remark. The introduction has been revised and improved by adding also more and updated bibliographical references, in order to answer the referee’s suggestions.

Overall, the presentation is OK and is recommended for a through language polishing and another round of review after the comments detailed below are addressed.

The authors would like to thank the anonymous referee for these suggestions. The paper has been revised and improved according to the referee’s comments, detailed below. Moreover, the English editing has been provided by the MDPI editorial office.

Abstract:

In general it should be edited to make the contribution of the paper more clear.

a)     L2; “offshore installation” it is two sided! You mean the installation operation of an offshore wind turbine or offshore wind turbine itself. Please make it clear.

The authors agree with the anonymous referee regarding this point. This issue has been fixed in the revised version of the paper.

b)     L1-2; Language: “The fault diagnosis of wind turbine systems represent a challenging issue, *, thus justifying the research topics developed in this work.”

The authors would like to thank the anonymous referee for this remark. This phrase has been modified in the revised version of the paper.

c)     L3; Typo: it represents

The authors would like to thank the anonymous referee for this remark. This typo has been modified in the revised version of the paper.

d)     L6; Language: “practical analytical knowledge” parallel structure adj+and+adj or adv+adj

Practical and analytical knowledge.

The authors would like to thank the anonymous referee for this remark. The occurrences of this form have been corrected in the revised version of the paper.

1-     Introduction

The introduction is really poor in terms of presented adequate and relevant references which lead to the niche of the paper. For example,

The authors would like to thank the anonymous referee for this remark. In the revised version of the paper, the literature review has been revised and enriched, as suggested.

a)     L36: Language and clarity. Like: “this will lead to prevent critical failures” change to “this would prevent….”

The authors would like to thank the anonymous referee for this remark. This point has been corrected in the revised version of the paper.

b)      L43,”[2,3]”: maybe better to cite review papers if you want to point general research in the field.

The authors would like to thank the anonymous referee for this remark. This issue has been fixed in the revised version of the paper, by including new bibliographical references.

c)     L44, lack of reference: “at a wind turbine level”

The authors would like to thank the anonymous referee for this remark. This point has been fixed in the revised version of the paper.

d)     L48-50, redundant and doesn’t convey useful information!

The authors would like to thank the anonymous referee for this remark. This suggestion has been included in the revised version of the paper.

e)     There are some language issues in the other paragraphs and the literature review is not systematically presented to direct the reader to the need of what has been done in the paper with proper referencing in each step. Please revise the section completely.

The authors would like to thank the anonymous referee for these suggestions, which have been considered while preparing the revised version of the manuscript.

2-     Offshore wind turbine simulator

The main issue in this section is that it refers to 3 different fault cases: sensor, actuator and system faults. However for the modelling only additive faults have been considered. While, the nature of change in the system dynamics is multiplicative faults so how to comment on that?

The authors would like to thank the anonymous referee for this remark. As remarked after Eq. (7), the faults are modelled as additive and equivalent input-output descriptions. This means that, according to the equivalent input and output injection principle, the actual effect of the wind turbine faults is described as if it were generated by equivalent signals added to the input and output measurements. This approach was formerly exploited for the fault diagnosis of gas turbines as described in:

Silvio Simani, Cesare Fantuzzi and Ron J. Patton, Model-based fault diagnosis in dynamic systems using identification techniques. Springer-Verlag Ed., ISBN: 1852336854. Advances in Industrial Control Series. London, UK. First Ed. November, 2002.

Moreover, this representation is also assumed for motivating the relations of Eq. (7) and Figure 2. Moreover, this description is also known as Errors-In-Variables (EIV) modelling, which is exploited in the dynamic system identification framework, as described in:

van Huffel, S., Lemmerling, P. (Eds.). Total Least Squares and Errors-in-Variables Modeling Analysis, Algorithms and Applications. Springer Verlag, 2002. Hardcover ISBN: 978-1-4020-0476-6.

This remark has been included in the revised version of the paper.

3. Intelligent Fault Diagnosis Techniques

Generally the language and typos should be revised. Like

a)     L127: Typo: “relying on on fuzzy system”.

The authors would like to thank the anonymous referee for this remark. This typo has been fixed in the revised version of the paper.

b)     L217, Language:“regards”

The authors would like to thank the anonymous referee for this remark. This point has been fixed in the revised version of the paper.

c)     How to derive the fuzzy rules? Any algorithm to obtained the optimal ones?

and didn’t comment on the choice of the fuzzy topology.

The authors would like to thank the anonymous referee for these remarks. These issues have been fixed in the revised version of the paper.

d)     Neural Network: why this architect of the network? Please comment of the training data and approach, testing data and number of neurons.

The authors would like to thank the anonymous referee for these remarks. A few more details have been included in the revised version of the paper.

e)     Table3: sensitivity fault analysis has been done by considering one fault at a time, yes? However, in reality there might be multiple faults at the same time with might have effect on the nature of the system and change the sensitivity, how to comment on this?

The authors would like to thank the anonymous referee for these remarks. A few more details have been included in the revised version of the paper.

f)      Table3: I am wondering due to symmetry, why, in Faults 1,2,3 is repeating and not for blade 2 or 3. And for the output why not the and not if they are redundant sensors.

The same issue for faults 6, 7.

The authors would like to thank the anonymous referee for these remarks. These issues depend on the structure of the wind turbine simulator, which was developed by kk-electronic for the international competition on FDI and FTC solutions, as recalled in the paper. Indeed, due to the symmetry, Faults 1, 2 and 3 are not repeated for blades 2 and 3. They would have the same effect, and it would not provide any useful information. On the other hand, a reduced number of sensors was implemented for measuring the most important output variables. However, as remarked in the works [4, 7], the structure of the simulator was motivated by a feasibility study regarding the development of FDI and FTC solutions. These remarks are included in the revised version of the paper.

4. Results and Discussion

Again language problem.

a) L303-305: Language, vague!

The authors would like to thank the anonymous referee for this remark. This part has been modified and removed in the revised version of the paper.

b) L293: please comment on the WT, rated power, wind speed …

The authors would like to thank the anonymous referee for these remarks. A few more details have been included in the revised version of the paper. On the other hand, since the anonymous referee #2 suggested to reduce the details describing the wind turbine simulator in Section 3, a few more references were included.

c) Table4: why fault case 7 is considered as trapezoidal while the fault case 9 is more of a nature of gradual change due to for example wear in drive train?

The authors would like to thank the anonymous referee for these remarks. The wind turbine simulator was developed by kk-electronic (Denmark) and motivated by the international competition on FDI and FTC, as referenced in [4, 7] of the revised version of the paper. This simulator implemented different fault cases, as described in [4]. In particular, the simulator includes more redundant sensors, which measure the same variables, such as the pitch blade angles (3) or the generator angular speed (2). However, the measurement sensors are affected by Gaussian noise that should describe the real measurement system. Therefore, noise processes with different standard deviations can affect different sensors measuring the same variables, such as the pitch blade angles (3). On the other hand, the fault case 7 is modelled with a trapezoidal function that is directly added to the corresponding output measurement. Differently, the fault case 9 is generated as a step change of the natural frequency and damping of the transfer function describing the drive-train model. Of course, the effect of this fault (case 9) on the output measurements is different from a step function. More details regarding the fault scenario can be found in [4, 7]. These comments are briefly summarised after Table 4, in the revised version of the paper.

d) Table 9: the presented results like FAR and MFR is really poor how to comment on it?

The authors would like to thank the anonymous referee for these remarks. However, the achieved results are very good indeed! First, Table 9 summarises the results achieved via a HIL test rig, i.e. in real time conditions. Moreover, the values are expressed over 1000 simulations, and not as percentage. Therefore, both FAR and MFR are equal or less than 0.5% (0.005). On the other hand, these values meet the limits required by the international competition [12].

Reviewer 2 Report

Dear authors,

Although the content of the article is scientifically interesting and in line with the latest trends in artificial intelligence applied to fault diagnosis, the structure of the paper can be significantly improved. In fact, the reading, in my opinion, is rather boring mainly because of the length of the article.

The main deal a non clear definition of the aim of the work.

There are therefore many sections that can be omitted or strongly shortened if you define properly the purpose of the research more clearly.

For example:

Paragraph 2.1 describes the physical model of the turbine; I would fully eliminate it by simply inserting a bibliographic reference to the model used (The model: did you develope it? ... is it validated?).

Paragraph 2.2 is really useful for the description of the search or the bibliographic references cited are sufficiently descriptive? I think so

Paragraphs 3, 3.1 and 3.2 ... is such a comprehensive treatment necessary?

Paragraph 3.3: I believe this is the core of the research, but, I repeat, more clarification must be given in this regard.

Results and discussion: again, the article is lacking in synthesis.

Conclusions: they simply summarize the global content of the article, without entering into the merits of the results obtained.

Moreover:

1. The introduction is not properly "centered" on the state of the art of the activity that you carried out, and it is poor in terms of literature review available about the "wind turbine fault diagnosis" ... only 6 papers!

2. I recommend submitting the new version of the article to a native english speaker since in many cases some expressions are not proper of the english.

For the reasons listed above I consider the article in its current form not suitable for publication and suggest a major revision.

Author Response

The paper has been revised according to the comments received by the two anonymous referees. All changes have been highlighted in the revised version of the paper and the response to reviewers’ comments is reported in the following.

The authors would like to thank the Managing Editor and the anonymous referee for her/his comments and suggestions, which were taken into account for improving the revised version of the manuscript. In the following, the anonymous referee’s comments and remarks are summarised (in black colour text) and followed by the authors’ answers, highlighted in blue colour. Moreover, the modified parts of the manuscript have been highlighted in yellow.

Anonymous Referee 2 Comments

Although the content of the article is scientifically interesting and in line with the latest trends in artificial intelligence applied to fault diagnosis, the structure of the paper can be significantly improved. In fact, the reading, in my opinion, is rather boring mainly because of the length of the article.

The authors would like to thank the anonymous referee for this remark. Section 1 has been improved and enriched, by adding more bibliographical references; on the other hand. Sections 2 and 3 reduced, in order to highlight the main contributions and novelties of the paper.

The main deal a non clear definition of the aim of the work.

The authors would like to thank the anonymous referee for this remark. In particular, Section 1 has been improved in order to highlight the novelty aspects of the proposed investigations. Moreover, redundant parts are omitted.

There are therefore many sections that can be omitted or strongly shortened if you define properly the purpose of the research more clearly.

For example:

Paragraph 2.1 describes the physical model of the turbine; I would fully eliminate it by simply inserting a bibliographic reference to the model used (The model: did you develope it? ... is it validated?).

The authors would like to thank the anonymous referee for this remark. Section 2 has been reduced, as more details regarding the wind turbine simulator can be easily found in the related literature. The model was developed by kk-electronic (Denmark) and MathWorks for the international competition on FDI and FTC solutions, as described in [4, 7]. More details on this point have been included in the revised version of the paper.

Paragraph 2.2 is really useful for the description of the search or the bibliographic references cited are sufficiently descriptive? I think so

The authors would like to thank the anonymous referee for this remark. Section 2 has been reduced, as more details can be easily found in the related literature.

Paragraphs 3, 3.1 and 3.2 ... is such a comprehensive treatment necessary?

The authors would like to thank the anonymous referee for this remark. Also Section 3 has been reduced, in order to highlight the most important aspects of the proposed study. Redundant parts have been removed.

Paragraph 3.3: I believe this is the core of the research, but, I repeat, more clarification must be given in this regard.

The authors would like to thank the anonymous referee for this suggestion. Section 3 has been improved, in order to highlight the most important aspects of the proposed study.

Results and discussion: again, the article is lacking in synthesis.

The authors would like to thank the anonymous referee for this remark. Section 3 has been revised, in order to highlight the most important points of the proposed study. Moreover, redundant parts have been removed from the revised version of the paper.

Conclusions: they simply summarize the global content of the article, without entering into the merits of the results obtained.

The authors would like to thank the anonymous referee for this remark. In general, Conclusion section should remain a stand-alone part of the manuscript, and it cannot contain too many details regarding the core of the paper. In fact, the discussion of the achieved results is usually drawn in the last part of the section addressing the simulations and experimental studies. However, without increasing the length of the Conclusion section, a few comments regarding the achieved results have been included.

Moreover:

1. The introduction is not properly "centered" on the state of the art of the activity that you carried out, and it is poor in terms of literature review available about the "wind turbine fault diagnosis" ... only 6 papers!

The authors would like to thank the anonymous referee for this remark. In particular, Section 1 has been improved in order to provide a sufficiently clear overview and critical discussion of the related literature. More bibliographical details have been included in the revised version of the paper.

2. I recommend submitting the new version of the article to a native english speaker since in many cases some expressions are not proper of the english.

The authors would like to thank the anonymous referee for this suggestion. The paper has been revised according also to the remarks provided by the Anonymous Referee #1. Moreover, the English of the paper was edited and checked by the editing service provided by MDPI.

For the reasons listed above I consider the article in its current form not suitable for publication and suggest a major revision.

The authors would like to thank the anonymous referee for her/his remarks, which have been carefully taken into account in order to definitely improve the quality of the revised paper.

Round 2

Reviewer 1 Report

The paper has been updated to account for the previous comments. I would recommend it for publication.